# Recent Progress in the Multicomponent Synthesis of Pyran Derivatives by Sustainable Catalysts under Green Conditions

**DOI:** 10.3390/molecules27196347

**Published:** 2022-09-26

**Authors:** Suresh Maddila, Nagaraju Kerru, Sreekantha Babu Jonnalagadda

**Affiliations:** 1Department of Chemistry, GITAM School of Sciences, GITAM University, Visakhapatnam 530045, Andhra Pradesh, India; 2School of Chemistry & Physics, University of KwaZulu-Natal, Westville Campus, Chiltern Hills, Durban 4000, South Africa; 3Department of Chemistry, GITAM School of Science, GITAM University, Bengaluru Campus, Bengaluru 561203, Karnataka, India

**Keywords:** green synthesis, single-step synthesis, pyrans, heterogeneous catalyst, multicomponent reactions

## Abstract

Pyrans are one of the most significant skeletons of oxygen-containing heterocyclic molecules, which exhibit a broad spectrum of medicinal applications and are constituents of diverse natural product analogues. Various biological applications of these pyran analogues contributed to the growth advances in these oxygen-containing molecules. Green one-pot methodologies for synthesising these heterocyclic molecules have received significant attention. This review focuses on the recent developments in synthesising pyran ring derivatives using reusable catalysts and emphasises the multicomponent reaction strategies using green protocols. The advantages of the catalysts in terms of yields, reaction conditions, and recyclability are discussed.

## 1. Introduction

Heterogeneous catalysis (HC) has become a valuable means for designing sustainable production protocols to obtain superior intermediates, synthons, and potent bioactive heterocycles [1,2,3]. The heterogeneous catalysts proved economical and eco-friendly due to the effectiveness of non-hazardous metals for the construction of nanomaterials for atom-economical organic reactions [4,5]. Heterogeneous catalysts significantly affected scientific and industrial innovations due to their tuneable textural properties, flexible acid–base characteristics, higher thermal stability, and improved reusability, with their excellent performance over their enormous counterparts [6,7,8,9,10,11]. Due to the enhanced utility in academia and industries, these materials have received considerable attention in recent years as reusable catalysts to improve the rate of chemical transformation with excellent yields and create substantial routes [12,13,14,15]. Over the years, diverse transition metal oxide nanomaterials have been investigated as sustainable, recyclable catalysts for developing bioactive promising small organic molecules [16,17,18]. Thus, the research outcomes have showcased the path for designing novel nanomaterials and investigating the mechanism and reactivity to allow the new ideal routes of organic synthesis [19,20,21].

The prominence of viable heterogeneous catalysts and economic advantages is being revived through efficient green protocols, which are considered vital in improved organic synthesis [21,22,23]. In particular, one-pot synthetic methodology protocols under green conditions of heterocyclic analogues have become a sizable alternative in modern organic chemistry to conventional multistep synthesis [24,25,26,27]. Because of the improved productivity and substantial time saving, multicomponent reactions (MCRs) are recognised as reliable synthetic green methodologies [21,28,29,30,31]. Several advantages of MCRs driven by green credentials are crucial in organic synthesis and drug discovery programs [32,33,34,35,36,37,38,39]. Hence, chemists focused on eco-friendly methods to build heterocyclic molecules under sustainable green protocols.

Oxygen-based heterocyclic frameworks have prominence in organic synthesis due to their significant pharmacological applications [40,41,42,43,44,45]. Among the various oxygen-heterocycles, pyran ring moieties have gotten immense consideration for a broad spectrum of medicinal properties. Antidiabetic, anti-HIV, antituberculosis, anticancer, antiproliferative, and antimicrobial activities are some of the prime assets [46,47,48,49,50,51]. Currently, various drugs involving the pyran skeleton are accessible in the market for biological applications (Figure 1). Based on the different therapeutic applications of the pyran ring skeleton, synthesising pyran analogues through MCR strategies by employing diverse heterogeneous catalysts received considerable attention [52,53]. This monograph summarises the advances in the one-pot construction of the molecules with pyran ring derivatives via MCR using sustainable heterogeneous catalysts under green protocols.

## 2. Synthesis of Pyran Derivatives

### 2.1. Iron Base Catalysts

Shabani and co-workers [54] concocted an iron-based chlorosilane hetero-nanocomposite and used them synthesised 4*H*-pyran analogues via green approaches. The prepared magnetic nanocatalyst was subjected to various spectral characterisations such as VSM, TGA, FT-IR, XRD, TEM, SEM, and AFM. VSM established its super magnetic properties, and its particle size was on an average of 23 nm in diameter, as suggested by SEM, TEM and AFM analysis. Its catalytic efficiency in preparing 4*H*-pyran derivatives (**4**) via Knoevenagel condensation of substituted aldehydes (**2**), active methylene group (**3**) and methyl acetoacetate (**1**) in ethanol was assessed. The catalytic amount of 10 mg was sufficient to gain a pyran yield of 97% in 15 min of reaction time (Figure 1).

Heravi and co-workers [55] have developed and studied the efficiency of magnetic nanocatalyst embedded with palladium (Fe_3_O_4_@SiO_2_-NH_2_@Pd). To formulate the catalyst, Fe_3_O_4_ nanoparticles were initially prepared by the co-precipitation method of iron sulphates in a 1:2 (Fe^2+^:Fe^3+^) ratio in the aqueous medium. The obtained Fe_3_O_4_@SiO_2_@PrNH_2_ were given an alcoholic wash and vacuum dried at 60 °C for one whole day. Palladium was embedded in the resulting magnetic nanocatalyst by mixing it with Pd(OAc)_2._ The magnetic nanocatalyst with palladium collected by magnetic decantation was washed with ethanol and vacuum dried. The prepared nanocatalysts were characterised using SEM, TEM, FTIR, TGA and EDS, which revealed a uniform spherical shape of the average particle size being 22.7 nm with a core diameter of 21 nm. The EDS data also collected the palladium embedment data, i.e., 2.74 × 10^−3^ mol/g, with certainty. Its catalytic activity was measured using a model reaction between substituted benzaldehyde (**4**), active methylene group (**3**) and dimedone (**5**)/ethyl acetoacetate (**1**) in a single-pot system. The reaction followed the Knoevenagel condensation mechanism to gain pyran (**7**) and teterahydro-chromene (**6**) derivatives (Figure 2). The catalyst was quite efficient, with a small quantity of 10 mg to achieve 98% product yield. The catalyst maintained its worth with any considerable degeneration for ten consecutive cycles.

Esmaeili and co-workers [56] formulated a magnetic biopolymer nanocatalyst using xanthan gum and used it to synthesise amino cyano-pyran analogues. The catalyst was prepared by dispersing iron chloride with an ionic ratio of 1:2 for Fe^2+^:Fe^3+^ in a solution containing xanthan gum (Fe_3_O_4_@Xanthan gum). The prepared catalyst was characterised via FT-IR, FE-SEM, EDX, VSM, TGA, and XRD. The analysis revealed that Fe_3_O_4_/xanthan gum had a magnetic concentration of 20 emu/g with monodispersed nanospheres of 14–38 nm with crosslinking among biopolymer matrixes. Its catalytic efficiency on the Knoevenagel condensation between substituted aldehydes (**8**), malononitrile (**3**) and dimedone (**5**) in ethanol was evaluated. The transient intermediate, α-cyanocinnamonitrile, further undergoes addition and tautomerisation to yield 2-amino-3-cyano-4*H*-pyran analogues (**9**) (Figure 3). The biopolymeric nanocatalyst, Fe_3_O_4_/xanthan gum, was successful in producing a yield of 96% within 4 min of reaction time at room temperature. The material did not lose its activity even after nine reaction cycles at a stretch.

Maleki and co-workers [57] have developed magnetic nanoparticles functionalised with SO_3_H dendrimers, which can be used as a hetero-catalyst (Fe_3_O_4_@-D-NH-(CH_2_)_4_-SO_3_H) for the synthesis of pyrans (**11**) and polyhydroquinolines (**14**). Structures established using TEM, FT-IR, SEM, TGA, EDX, and XRD suggested well-established dendrimeric bondage with Fe_3_O_4_, uniform spherical formats with a dark magnetic core, and a grey matrix of dendrimer units. It proved an ideal catalyst for the reaction between aldehyde (**10**), melanonitrile (**3**) and dimedone (**5**) and cyclohexanedione (**13**) (Figure 4). The proposed mechanism was via Knoevenagel condensation of malononitrile to aldehyde leading to the formation of benzylidenemalononitrile. The product further undergoes addition with cyclohexanedione via Michael addition and cyclisation, resulting in 4*H*-pyran scaffolds. The prepared nanocomposites offered excellent yields, up to 92% within 10 min, using 0.5 g of the catalyst. The catalyst also retained its capability for five consecutive cycles without any degradation.

Mohammadi and Sheibani [58] described the magnetic nanocomposites of guanidine poly acrylic acid functionalised with SO_2_ (Fe_3_O_4_@SiO_2_-guanidine-PAA). The prepared magnetic nanoparticles were characterised using VSM, TGA, FTIR, TEM, SEM and EDS analysis which suggest a central amorphous shell of silica encapsulating Fe_3_O_4_ with a diameter of 14 nm. The morphological size of Fe_3_O_4_/SiO_2_-guanidine@Poly acrylic acid was 20 nm in diameter. Its catalytic activity in preparing dihydro pyranochromenes (**17**) and tetrahydro benzopyran (**18**) moieties were evaluated. The one-pot system at 70 °C involved malononitrile (**3**) and aromatic aldehyde (**15**). In addition, hydrocoumarin (**16**) in the former reaction and dimedone for the second (Figure 5). The catalyst produced excellent yields of 98% in 20 min of reaction time with a minimalist utilise of 50 mg of the catalytic load.

Jamshidi et al. [59] have tailored magnetic nanoparticles adhered with HPA dendrimer, a hybrid catalyst material for synthesising pyran analogues (**20** and **21**), via a green protocol. The prepared nanocomposites were characterised using various analytical techniques such as FTIR, TGA, SEM, XRD and VSM, which suggest a sphere-shaped nanoparticle ranging from 23–25 nm in diameter. A single-pot system involving malononitrile (**3**), dimedone (**5**) or ethyl acetoacetate (**1**) and various benzaldehydes (**19**) (with electron-donating and withdrawing groups) in the ethanolic medium under reflux conditions were examined (Figure 6). The catalyst produced a 92% yield within 5 min of reaction time. The reaction mechanism was assumed to proceed via Knoevenagel condensation, Michael addition and cyclisation in sequence. The catalytic activity originated from the conjugated nanocomplex, Fe_3_O_4_@Dendrimer-NH_2_-HPA, by a leaching arrangement. The catalyst excels in various aspects such as easy separation, environmentally supportive and reusability for several reaction cycles without any degradation.

Elhamifar and co-workers [60] prepared phenylsulphonic acid affiliated magnetite nanoparticles. They employed them as reusable catalysts for preparing tetrahydropyran (**23**) analogues. The collected nanoparticles of Fe_3_O_4_@PhSO_3_H were subjected to acidic determination through ion-exchange pH assessment, which was reflected in 1.6 mmol/g of sulphonic acid. It was characterised by different analytical instruments such as PXRD, TGA, FTIR, SEM, TEM, VSM and EDX. The thermal stability of the catalyst was below 100 °C. Its sulphonyl and phenyl groups were removed at 400 °C and 700 °C, respectively. The average particle size was 25 nm of spherical particles with a magnetic core and a matrix of phenylsulphonic acid. The catalytic performance was evaluated using a MCR consisting of benzaldehyde (**22**), malononitrile (**3**) and dimedone (**5**) in the presence of 0.2 mg of Fe_3_O_4_@PhSO_3_H nanoparticles under ultrasonication (Figure 7). The catalyst produced a product of 95% within 25 min of the reaction time. The catalyst was stable and durable for nine consecutive cycles with the same affinity.

Aghbash and co-workers [61] have tailored magnetic silica functionalised with MCM-41 adhered to DABCO (Fe_3_O_4_@silica-MCM-41@DABCO) and used as a catalyst material for synthesising amino dihydropyrano analogues (**26**). The catalyst was characterised using different analytical methods such as VAM, XRD, FTIR, SEM, EDX and TEM. A mesoporous framework with a regular spherical morphology of a particle size 19–67 nm and magnetisation of 12.5 emu/g was suggested. The catalytic affinity of the prepared nanocatalyst was evaluated in the preparation of amino dihydropyrano cyanoparan moieties using a multicomponent system of benzaldehyde (**25**), azido-KA (**24**) and malononitrile (**3**) in DMF solvent medium. The catalyst produced excellent yields up to 98% with a catalytic load of 3 mg under reflux within half an hour (Figure 8). The mechanism of the product formation was presumed to proceed through Knoevenagel condensation leading to Michael’s addition. This method has an advantage in getting high yields at shorter reaction spans.

Mofrad and co-workers [62] have developed silica-adhered magnetite nanoparticles composed of ferrocene-infused ionic liquid, which can be used as green catalytic media for the preparation of amino cyano pyran (**29**) scaffolds. The authors established the catalyst characteristics using FTIR, SEM, XRD, TEM and EDX techniques. The magnetisation of the prepared particles was 28 emu/g, and a spherical shape with a magnetic core enveloped in a silica shell with a matric of imidoferrocene having a crystalline structure of 50.39 nm in diameter. To assess the catalytic efficiency of [Fe_3_O_4_/SiO_2_/Im-Fc][OAc], a single-pot system consisting of β-naphthol (**27**), malononitrile (**3**) along with an aldehyde (**28**) was used. In the absence of solvent, the mixture was continuously stirred at 90 °C to obtain the desired amino cyano pyran moieties (Figure 9). The feasible reaction mechanism proceeded via Knoevenagel, Michael and tautomeric cyclisations. The catalyst with a load of 10 mg generated the product of 94% yield within 15 min. This catalyst offers high yields with a short reaction time with minimal usage.

Manesh and co-workers [63] have developed magnetic nanoparticles of Fe_3_O_4_/SiO_2_/CPS encapsulated with spiro[indoline-3,4′-[1,3][dithiine]-Ni-composite, which can be used as green heterogeneous catalyst for the preparation of dihydropyran (**32**) scaffolds. The magnetic nanoparticles were characterised using FTIR, XRD, EDX, VSM, BET, SEM and TEM. The material possessed a mesoporous particle size of 16 nm and a surface area of 16.92 m^2^/g with 55.1 emu/g of magnetisation. The catalytic properties of the catalyst were confirmed using a single-pot reaction system consisting of oxirane (**31**) and isocyanide (**30**) in ethanol followed by malononitrile (**3**) with continuous stirring at 60 °C (Figure 10). The reaction was carried out in a basic medium from which the desired pyran product was separated from the catalyst using an external magnetic field. The prepared pyran moieties were washed and dried over MgSO_4._ The pyran moieties were characterised using various spectral analyses such as FTIR, ^1^HNMR and ^13^CNMR. The catalyst resulted in good yields of 88% within 8 h of reaction time and a catalytic load of 15 min under reflux. The proposed mechanism leads via Knoevenagel, Michael addition followed by cyclisation.

Maleki and co-workers [64] synthesised tetrahydro pyran (**34**) scaffolds using a Schiff-nickel complex and infused them with magnetite nanoparticles (Fe_3_O_4_@SiO_2_@NiSB). The gathered catalytic nanoparticles were characterised using FTIR, SEM, EDX, XRD, TEM, TGA and VSM. The nanoparticles were spherical with 35 nm in diameter and a surface area of 61.606 m^2^/g. The catalytic capacity was evaluated through an MCR consisting of benzaldehyde (**33**), malononitrile (**3**) and dimedone (**5**) to prepare tetrahydro pyrans in a solvent-free environment (Figure 11). The catalyst yielded excellent products of up to 98% with a catalytic load of 10 mg within a reaction time of 5 min at room temperature. The reaction occurs via Knoevenagel condensation and Michael’s addition. The catalyst produced excellent yields and considerably decreased the reaction time under green optimisation.

Dazmiri and co-workers [65] have extracted magnetic zinc oxide nanoparticles (Fe_3_O_4_/ZnO MCNPs) from the water extract of *Petasites hybridus rhizome* to be used as an eco-friendly catalyst for the synthesis of pyran scaffolds. They also studied the antimicrobial and antioxidative properties of the prepared pyran scaffolds (**36**). The gathered nanoparticles were characterised using FTIR, EDX, XRD, SEM and TEM, which determine their particle size to be 40 nm in diameter with a uniform distribution of elements. The catalytic activity was evaluated using a single-pot reaction consisting of aromatic aldehyde (**35**), dimedone (**5**), malononitrile (**3**) and catalyst composite in an ethanolic medium. The obtained pyran yield was excellent, up to 90% within 10 min at room temperature (Figure 12). The reaction mechanism proceed via Knoevenagel and Michael’s reactions. The antimicrobial properties were tested against E. Coli bacteria, whereas the antioxidant and reducing properties were evaluated using DPPH and Fe^3+^, respectively.

Afruzi and co-workers [66] have developed a polymer of magnetised acrylonitrile in combination with melamine (PAN@melamine/Fe_3_O_4_) that can be used in the preparation of pyrano pyrazoles (**40**) and amino cyano pyrans (**39**). The gathered magnetic polymeric chains were characterised using FTIR, XRD, EDX, TGA, SEM, TEM and VSM, which suggest a spherical shape with 40 nm in diameter and magnetisation of 29.2 emu/g. A single-pot reaction consisting of benzaldehyde (**37**), malononitrile (**3**), hydrazine hydrate (**38**) and acetoacetate (**1**) in an ethanolic medium under the influence of the polymer material (Figure 13) was evaluated. The reaction was proposed as a combination of Knoevenagel and Michael and tautomeric cyclisations. The catalyst load of 10 mg was efficient in generating 97% of yield in 10 min. The catalyst did not show any considerable degradation even after five consecutive cycles and performed almost as a freshly prepared catalyst for every use. This method’s assets are eco-friendly conditions and easily recyclable catalysts.

Heravi et al. [67] synthesised pyran (**44**) and pyrimidine (**45**) analogues using magnetite nanoparticles (nano-Fe_3_O_4_) as a heterogenous catalyst. The prepared magnetic nanoparticles were characterised using FTIR, SEM, TEM, and VSM, which suggest a spherical shape of the catalyst particles. A mixture consisting of benzaldehyde (**41**), malononitrile (**3**) and pyruvic acid (**42**) was used as a model reaction in the presence of magnetite nanoparticles (Figure 14). They also compared various other catalysts under the same reaction conditions. The freshly prepared magnetite nanoparticles produced a 91% yield within a reaction span of one hour under reflux with a catalytic load of 20 mg. The target compounds were analysed using FTIR, ^1^H and ^13^C NMR. This catalyst was rotated for four consecutive cycles without considerable degradation. The probable mechanism involves Knoevenagel reaction and magnetite acting as Lewis acid to facilitate the Michael addition and cyclisation.

Sandaroos et al. [68] have synthesised tetrahydro furopyran (**48**) scaffolds using iron triflate nanocatalyst (Fe(OTf)_3_). It was explored as a catalyst for MCR system consisting of aromatic aldehyde (**46**), tetronic acid (**47**) and malononitrile (**3**) in the presence of acetonitrile. The reaction mixture was refluxed at ambient temperature with continuous stirring to obtain the desired furopyran moieties separated from the solvent through evaporation and then diluted with CH_2_Cl_2_ and water (Figure 15). The reaction mechanism was expected to proceed through Knoevenagel and Michael reaction within a reaction time of 7 h to produce a yield of 92% with a catalytic load of 15 mg. This method has the advantage of excellent harvests with an easy workup plan.

### 2.2. Zinc-Based Catalysts

Ziyaadini and co-workers [69] have fabricated catalytic composites of Zn_2_SnO_4_/SnO_2_ using the sol-gel method. The catalyst material was characterised via XRD, TEM and SEM, which showed that it possesses two cubic and a tetragonal structure in apropos to Zn_2_SnO_4_/SnO_2_ nanocomposites. The catalytic efficacy of the nanocomposites was examined for the preparation of pyrano [2,3-c]-chromene derivatives (**50**) through multicomponent strategy. They devised a multicomponent reaction system using 4-hydroxy-chromenone (**16**), substituted aldehydes (**49**) and active methylene compound (**3**) (Figure 16). The reaction proceeds in the presence of the formerly prepared Zn_2_SnO_4_/SnO_2_ nanocomposites in ethanol under ultrasonic irradiation at 80 °C. This reaction was presumed to be a two-step process that starts with Knoevenagel condensation followed by the cyclisation step, which leads to the desired analogue. A 0.05 g of the catalyst was quite efficient in producing a 92% yield. Aldehydes with electron donating or withdrawing substitutes offered a better yield of pyranochromene derivatives. The catalyst was reused effectively for eight successive runs.

Maleki and co-workers [70] have fabricated a nanocomposite consisting of alginic acid enwrapped onto ZnFe_2_O_4,_ which acts as an effective green catalyst in the preparation of amino cyano pyran scaffolds (**52**). The magnetic core was prepared through co-precipitation of Fe(NO_3_)_3_ and Zn(NO_3_)_2_ at alkaline pH and dispersion of alginic acid under ultrasonication at 80° for 40 min. The catalyst was characterised using FTIR, SEM, TEM, VSM, ICP-AES, EDX and XRD, which indicated 70 nm sphericle nanoparticles and an average crystalline size of 27 nm. The catalytic efficiency was evaluated using a single-pot system of malononitrile (**3**), dimedone (**5**) and aromatic aldehyde (**51**) (Figure 17). The reaction was presumed to proceed through Knoevenagel, Michael addition and intramolecular tautomeric cyclisation. The nanocatalyst, ZnFe_2_O_4_/alginic acid, was efficient in producing excellent yields of amino cyano pyrans of approximately 83–95% with 5 mg catalyst load at room temperature in 15 min. The catalyst material was reusable for five successive runs without losing its abilities.

Tahmassebi et al. [71] have studied the efficiency of the Zn(Proline)_2_ complex in the synthesis of annulated pyran derivatives (**55**). In a single pot, the mixture of methyl phenyl pyrazolinone (**54**), aromatic aldehyde (**53**) and malononitrile (**3**) was stirred in the ethanolic medium for 3 h under reflux conditions (Figure 18) in the presence of the catalyst. The reaction was assumed to proceed through Knoevenagel condensation and Michael addition, followed by tautomeric cyclisation and conjugation. The use of Zn (Proline)_2_ in this reaction produced excellent yields of 94% within a reaction time of 3 h and a catalytic load of 10 mg. This catalyst is advantageous regarding water solubility and a clean reaction profile within a short time.

Khan et al. [72] have synthesised pyran analogues (**58**) using ZnCl_2_ as a heterogeneous catalyst. A single-pot multicomponent reaction system consisting of *N*-methyl thio nitroetheneamine (NMSM) (**57**), β ketoester (**1**) and aromatic aldehydes (**56**) in the presence of zinc chloride was used as a model reaction (Figure 19). The reaction proceeds in a no-solvent environment at 120 °C. The reaction pathway was presumed to follow Knoevenagel condensation along with Michael addition and cyclisation to obtain the respective pyran derivatives. A product yield of 70% was observed within a reaction time of 30 min in a neat reaction environment with a catalytic load of 30 mg. This method is appreciated in terms of a green background and a short reaction time with satisfactory results. The gathered pyran derivatives were characterised using various spectral analytical techniques such as FTIR, single-crystal XRD, ^1^H and ^13^C NMR.

### 2.3. Silica-Based Catalysts

Ghomi and co-workers [73] have developed silica nanoparticles (SnCl_2_@nano SiO_2_) affixed with stannous chloride, which was used as a hetero catalyst for the preparation of tetrahydropyran (**60**) scaffolds. Silica gel nanoparticles were dispersed in dichloromethane, and stannous chloride was added to it while continuously stirring overnight at room temperature. The gathered nanoparticles were characterised by FTIR, SEM and EDAX, which reflect a 20 nm average particle size of the catalyst. The catalyst efficacy was tested for synthesising pyran moieties using a three-component system involving malononitrile (**3**), ethyl acetoacetate (**1**) and aryl aldehyde (**59**). Poor product percentages were observed in THF and DCM solvents, whereas protic solvents such as ethanol obtained 96% yield with a catalytic load of 15 mg in a reaction time of 25 min (Figure 20). The catalyst was separated from the mixture using ethanol and reused for eight consecutive cycles with similar efficiency.

Yaghoubi and Dekamin [74] have developed an isocyanurate framework of organosilica nanoparticles with periodic mesoporosity (PMO-ICS), which can be used in the preparation of tetrahydropyran moieties (**62**). A diverse range of amino cyano pyran molecules was prepared using the new catalyst. The catalyst was characterised using different instrumental performances such as FTIR, TGA, BET, TEM, EDX and SEM, representing an isothermal pattern of hydrogen hysteric loops. The catalytic surface area and pore volumes were 570 m^2^/g and 0.57 cm^3^/g, respectively. The efficacy of the nanoparticles was assessed using a multicomponent system consisting of an aldehyde (**61**) and dimedone (**5**) along with malononitrile (**3**) in an ethanolic medium (Figure 21). As the catalyst consisted of a Brønsted base and a Lewis acid moiety, it can be considered a bifunctional catalyst. The plausible mechanism was Knoevenagel, Michael, and cyclisation reactions. The prepared catalyst produced a product yield of 97% using a catalytic load of 10 mg under reflux within a reaction span of 5 min and recyclability for up to five consecutive cycles. This catalyst proves promising in terms of shorter reaction time, high yields, and low loading amount.

Sameri and co-workers [75] have tailored zinc-embedded Schiff’s base anchored onto CaO/SiO_2_ (CaO@SiO_2_-NH_2_-Sal-Zn) to be used as a reusable solid material for the synthesis of pyran analogues (**64**). The collected nanoparticles were characterised by different instrumental techniques such as FTIR, XRD, TGA, ICP-MS, EDS, SEM and TEM, which suggest a core-shell of CaO enwrapped in SiO_2_ with Zn doped Schiff’s matrix having an average particle size of 35 nm. The catalyst efficacy was evaluated via an MCR involving benzaldehyde (**63**), hydroxycoumarin (**16**) and malononitrile (**3**) in a solvent-free environment at room temperature (Figure 22). The proposed reaction mechanism involved Knoevenagel, Michael addition, along with cyclisation. This catalyst was efficient in producing 97% yield in 10 min with a catalytic load of 10 mg. The catalyst ran six consecutive cycles without any degradation of the performance.

Patel and co-workers [76] have synthesised pyrano pyrazole carbonitriles (**66**) using silica nanoparticles (nano-SiO_2_) generated from agro-waste. The nanosilica particles were subjected to characterisation utilising different instrumental techniques such as FTIR, XRD, SEM, BET, and TEM, which suggest an amorphous structure size having (100–200 nm) with a surface area of 215.6m^2^/g along with a pore volume of 0.269 cm^3^/g and diameter of 7.021 nm. The catalytic performance of the catalyst was assessed with a multicomponent reaction system in a single-pot consisting of benzaldehyde (**65**), ethyl acetoacetate (**1**), malononitrile (**3**) and hydrazine hydrate (**38**) in an aqueous media (Figure 23). The reaction proceeds via Knoevenagel and Michael additions leading to the formation of pyrano pyrazole carbonitriles. The catalyst produced a yield of 94% within 30 min at 80 °C using 10 mg of catalyst. The separated catalyst was used in five consecutive cycles without compromising its performance.

### 2.4. Carbon Nanotube-Based Catalysts

Adibian and co-workers [77] developed the MWMCNT infused with NH_2_ and functionalised with *N*-butylsulphonate (MWCNTs-D-(CH_2_)_4_-SO_3_H) to be used as a hetero-catalyst for the synthesis of tetrahydrobenzo-pyran and dihydroindeno-pyridine moieties. Initially, carboxylated MWCNTs were magnetised utilising a 1:2 ratio of Fe^2+^/Fe^3+^ while constantly stirring at 50 °C and maintaining an alkaline pH using ammonium solution. The carbon nanotubes were subjected to characterisation via FTIR, TGA, SEM, TEM and XRD, which revealed the diameter of functionalised MWCNTs to be in the range of 5–21 nm with an average diameter of 10 nm and a dendrimer width of 2–5 nm. The matrix of the CNTs filled with butylsulphonate was approximately 9 nm in thickness. The scope of the catalyst was evaluated using a multicomponent reaction between aldehydes (**67**), activated methylene compounds (**3**) and dicarbonyls (**5** and **69**) to generate pyrans (**68**) and indanedione pyridines (**70**) (Figure 24). The catalyst yielded > 92% of the products with 60 mg of the catalyst in 12 min. The catalyst was used for five consecutive cycles without any degradation.

Hojati et al. [78] have developed magnetic carbon nanotubes functionalised with polypyrrole and used it as a catalyst to synthesise tetrahydro pyran (**72**) analogues. The carbon nanotubes were functionalised with a carboxylic acid group by dispersing CNT in an aqueous solution containing HNO_3_ and subjected to ultrasonication for 30 min. The catalytic nanoparticles (PPY/Fe_3_O_4_/CNT) were characterised using FTIR, TGA, SEM and TEM to estimate the morphology of the catalyst to be a magnetic core of CNTs and a matric of polypyrrole and an average size of the particles to be 30 nm with excellent thermal stability. The catalytic affinity of the prepared CNTs was evaluated using a single-pot reaction consisting of aryl aldehyde (**71**), β-ketone (**5**) and malononitrile (**3**) in the presence of magnetic CNTs-PPY (Figure 25). The functionalised polymeric magnetic CNTs proved excellent by yielding 95% of pyran scaffolds at 90 °C with a catalytic load of 32 mg within a reaction time of 10 min. This procedure has an easy workup plan, minimal catalytic load, and excellent yields. The reaction mechanism is associated with Knoevenagel condensation followed by Michael addition and cyclisation.

Hajipour and Khorsandi [79] have studied the pros and cons involved in the usage of CNTs embedded with proline (Pro/MWCNTs) and proline being used as an organic ionic solvent catalyst in the preparation of amino pyran derivatives (**75**). The CNTs with proline suffixes were prepared using a multistep synthetic process where CNTs were infused with aniline via NaNH_2_ and H_2_SO_4_ at 60 °C which was later refluxed with DMF in nitrogen environment in the presence of proline to obtain the desired embedment whereas [BMIm][Pro] was considered as the desired ionic liquid catalyst. To assess the catalytic proficiency of the catalysts, a single-pot system consisting of aldehydes (**73**) and acetophenones (**74**) along with maloninitrile (**3**) in an ethanolic medium was used as a model reaction (Figure 26). CNT/Pro proved to be an excellent heterogeneous catalyst with incredible yields of up to 97%. The [BMIm][Pro] ionic liquid was considered an inexpensive and readily available catalytic medium. Both catalysts were recoverable and recyclable, yet CNT/Pro was better than [BMIm][Pro] in terms of reaction span.

### 2.5. Manganese-Based Catalysts

Norouzi et al. [80] developed a mesoporous ethyl-based organosilica magnetic nanocomposite-assisted ionic liquid infusing a manganese complex. The structure of the prepared nanocomposites (Mag/BPMO-Mn) was elucidated by employing FTIR, EDX, SEM, VSM, TEM and PXRD techniques. The analytical data revealed a magnetic core with a grey mesoporous organosilica shell having a spherical shape with an average size of 95 nm and a magnetisation value of 24.1 emu g^−1^. The catalytic efficiency of Mag/BPMO-Mn nanoparticles was explored to prepare 4*H*-pyran analogues using a reaction containing substituted aldehydes (**76**) and dimedone (**5**) along with malononitrile (**3**) in an aqua medium. The reaction mixture was continuously stirred at room temperature to afford 4*H*-pyran derivatives (**77**) (Figure 27). The excellent yield of 94% with 0.8 mol% of the nanocatalyst is attributed to its high surface area and isolated catalytic centres, and mesoporosity. The catalyst is easily separable and reused for several consecutive cycles, losing its productivity.

Maddila et al. [81] infused MnO_2_ onto hydroxyapatite heterogeneous catalyst (MnO_2_/HAp) for the preparation of pyran-carboxamide analogues (**80**). The catalyst was prepared using a co-precipitation method and subjected to characterisation via FTIR, p-XRD, SEM, and TEM, representing unequal cubes of 23 nm coagulated into oval-shaped black crystalline particles with a diameter ranging from 45–59 nm. Its catalytic properties were assessed using a single-pot system consisting of malononitrile (**3**), acetoacetanilide (**79**) and a substituted aromatic aldehyde (**78**) in one equivalent each (Figure 28). Aprotic mediums such as hexane rendered no reaction, and polar solvents such as THF, CH_3_CN, dioxane, DCM resulted in low yields. 50 mg of 3% MnO_2_ doped on hydroxyapatite offered excellent yields (89–98%) in H_2_O, ethanolic and methanolic media in 15 min reaction time. The feasible reaction mechanism was via Knoevenagel, Michael, and cyclisation reactions.

Mozafari and co-workers [82] have tailored a phosphotungstenic acid-aided SiO_2_-NHPhNH_2_ of MnFe_2_O_4,_ which can be used as an environmentally sustainable heterogeneous catalyst for the synthesis of tetrahydro benzopyran (**82**) and indazolo phthalazinetrione (**84**) moieties. The gathered nanoparticles were characterised using FTIR, which suggests the presence of phenylenediamine group on the surface of the MnFe_2_O_4_ nanoparticles. The other analytical techniques used include SEM, XRD, EDX, TEM, TGA, VSM and ICP-AES. The results indicate a magnetic spherical core covered with phenylenediamine matrix ranging from 26–50 nm in diameter and a surface area of 19.64 m^2^/g along with a pore size of 29 nm and a pore volume of 6.99 cm^3^/g. A single-pot system with benzaldehyde (**81**), dimedone (**5**) and phthalhydrazide (**83**) for phthalazinetrione moieties and malononitrile (**3**) for pyran moieties was used to establish the efficiency of the prepared nanocatalyst complex in an aqueous medium at 80 °C under sonication (Figure 29). The catalyst facilitated a 94% product yield within 8–10 min. The reaction is presumed to follow Knoevenagel condensation, nucleophilic addition and cyclisation of the formed intermediate to yield the target product.

### 2.6. Cobalt-Based Catalysts

Patel et al. [83] have developed self-assembled nanoflakes of Co_3_O_4_ as catalysts for the generation of tetrahydro benzopyrans (**86**) and 2-aryl 1,3-benzothiazoles (**88**). The catalyst was initially prepared through a hydrothermally induced co-precipitation process. The prepared catalyst was characterised using various analytical and spectral methods such as UV-Vis, photoluminescence spectra, powder XRD, SEM, EXD. The results suggest that Co_3_O_4_ is a pure poly-crystalline cubic spinel phase with crystalline dimensions of 10.28 nm and 26.71 nm and an average flake size of 145 nm. The catalyst efficacy of the Co_3_O_4_ was established via a one-pot system to generate tetrahydrobenzo pyran and aryl benzothiazole analogues. The reaction of substituted aldehydes (**85**), dimedone (**5**), malononitrile (**3**) and 2-aminothiophenol (**87**) in the presence of the catalyst (10 mg) in an ethanolic medium (1:1) offered good yields of 85% within a reaction span of 10 min (Figure 30). Further, the catalytic flakes sustained efficiency for ten consecutive cycles.

Tamoradi et al. [84] tailored spinel ferrites with Eu(III) hybrid nanocomposites (Eu-IDA@CPTS-CoFe_2_O_4_) and used them as catalysts in preparing tetrahydro benzopyran analogues (**90**). The prepared catalyst was subjected to characterisation via FTIR, SEM, TEM, EDX, XRD, WDX, which reveals a core concentrated with Co, Fe, and O in comparison with Eu, C, and N, thus, evidencing a core of CoFe_2_O_4_ with a matrix of organic Eu layer. The nanocomposite catalyst’s efficacy was evaluated with a one-pot reaction between substituted benzaldehyde (**89**), and activated methylene compounds such as malononitrile (**3**) and dimedone (**5**) (Figure 31). The presumed reaction pathway includes the Knoevenagel reaction followed by Michael addition and cyclisation. The catalyst (30 mg) offered a 96%yield within 25 min in an ethanolic medium at room temperature. The catalyst was used for six consecutive cycles with no loss of activity.

### 2.7. Silver-Based Catalysts

Tamimi and co-workers [85] have studied the use of the silver complex of phosphorous molybdate (Ag_3_[PMo_12_O_40_]) in the preparation of tetrahydro-pyran (**92**) moieties under green conditions. The catalyst was prepared by dissolving silver nitrate into an aqueous medium with continuous stirring at room temperature, to which H_3_[PMo_12_O_40_] was added dropwise till the yellow precipitate was separated. The catalytic particles were washed with ether and vacuum dried. The recovered material was again characterised using ICP, FT-IR, and hydrogen elemental analysis. Its catalytic efficiency was confirmed by employing an MCR involving aryl aldehyde (**94**), hydroxycoumarin (**16**) and malononitrile (**3**). The catalyst complex generated a 97% product yield in 60 min with a catalytic load of 0.05 g under reflux conditions (Figure 32). The reaction leads via Knoevenagel condensation and Michael addition. This method proved efficient for preparing pyran derivatives with high thermal stability and excellent yields.

Tavakoli et al. [86] have fabricated bimetallic coper and silver mesoporous (Ag@CuO@MCM-48) and used as a green catalyst for the preparation of pyran-pyrrole (**95**) combinations. The prepared catalyst was characterised using various methods such as FTIR, EDX, XRD, BET, SEM and TEM analysis, which confirmed the mesoporous spherical shape of the catalyst with a surface area of 73.8 m^2^/g and a particle size of 200 nm. The pore size of the particles was described as 3.7 nm with a total pore volume of 0.069 cm^3^/g. The activity of the bimetallic catalyst was evaluated using a one-pot system consisting of benzaldehyde (**93**), malononitrile (**3**) and methyl pyrrol oxopropanenitrile (**94**) in an ethanolic medium at 60 °C (Figure 33). The process produced pyran-pyrrol hybrid moieties (**95**) with excellent yields of 97% within 5 min of reaction time with a catalytic load of 25 mg. The proposed reaction leads through Knoevenagel condensation, Michael addition and cyclisation. This bimetallic catalyst provided excellent yields in a quick response and is recyclable for five consecutive cycles with similar activity.

### 2.8. Ionic Liquid-Based Catalysts

Moshtaghin and Abbasinohoji [87] have synthesised tetrahydropyran scaffolds (**97**) in a solvent-free environment using ionic liquid promoted by LaMnO_3_ (BeW_12_O_40_@ILMLMNPs). Initially, a kegging model salt with potassium in the ionic sphere was prepared through the hydrothermal method. The magnetic lanthanum nanoparticles were subjected to characterisation using FTIR, XRD, SEM and EDX data, which confirmed the Keggin-type polymeric spherical structure with a size of 40–90 nm. Its catalytic efficiency was established by the fusion of substituted aldehydes (**96**) with dimedone (**5**) and malononitrile (**3**) in the presence of the catalyst under solvent-free conditions. The mechanism involved Knoevenagel condensation, Michael addition and cyclisation (Figure 34). An excellent 95% yield within 10 min with 0.03 g of the catalyst at 80 °C was observed. The material retained its activity for five consecutive cycles.

Azra Zarei and co-workers [88] have synthesised bifunctional oxyammonium ionic liquid as a green and reusable catalyst for preparing pyranopyran analogues (**100**). To prepare the bifunctional catalytic liquid, firstly, diamino dioxoctone was added with trifluoroacetic acid dropwise under continuous stirring for 2 h. The material was further washed with diethyl ether to afford the desired catalytic liquid. The tagged ionic liquid was characterised using various spectral techniques such as FTIR, ^1^HNMR. ^13^CNMR, ^19^FNMR, COSY-NMR, TGA, DTG, XRD, SEM and TEM. A multicomponent reaction system consisting of malononitrile (**3**), aromatic aldehydes (**98**) and hydroxy methyl pyranone (**99**) in the presence of the ionic liquid was designed (Figure 35). Pyran derivatives with excellent yields of 96% were produced within a reaction span of 13 min at 80 °C with a catalytic load of 7 mg under solvent-free conditions. The catalyst was evaporated and collected for further use. The liquid could be reused for three cycles without significant degradation of activity.

Hashjin and co-workers [89] derived pyran scaffolds (**102**) using magnetic nanoparticles of zirconium impregnated ionic liquid (Zr@IL-Fe_3_O_4_). The collected catalytic nanoparticles were characterised via FTIR, SEM, EDX, TEM, VSM, and XRD analysis, indicating a circular morphology with a 23–35 nm diameter and a magnetic core having 48.40 emu/g magnetisation. The catalytic efficacy was established via a single-pot mixture consisting of benzaldehyde (**101**), malononitrile (**3**) along with hydroxycoumarin (**16**) to obtain pyran moieties (Figure 36). The reaction mechanism is predicted to proceed through Knoevenagel condensation followed by Michael addition and cyclisation. The protocol could produce the desired pyran moieties with a 96% yield in 15 min with a catalytic load of 10 mg at 100 °C. The catalytic activity was sustained for six consecutive cycles. This method adds an advantage with an easy workup plan to give the product excellent yields within a short reaction time.

### 2.9. Miscellaneous Catalysts

Kaminwar and co-workers [90] have suggested an updated pathway for the preparation of pyrano-[3,2-c]-quinoline (**105**) and dihydro-pyrano-[3,2-c]-pyran (**106**) derivatives via MCR system utilising copper nanocatalyst adhered to carbon microspheres (Cu-Np/C). The pyran and quinolines were prepared by Knoevenagel condensation of substituted aldehydes (**103**), diones (**104**/**99**) and active methylene compounds (**3**) in the presence of a water-ethanolic medium (1:1) at 80 °C (Figure 37). The reaction mechanism was assumed to be initiated with the oxidation of the surface copper particles to form CuO_2,_ which performs as a Lewis acid to activate the Knoevenagel condensation, followed by Michael addition and cyclisation of the intermediate formed. An excellent yield of 93% was obtained using 10 wt% of the prepared copper nanocatalyst. The catalyst was easily separated and used for several cycles without significant loss in its efficiency. Excellent yield and mild reaction conditions are the assets of this reaction method.

Abyar and co-workers [91] have synthesised 2-amino-[4,3-b]-pyranone derivatives by using titanium chloride bound to Kaolin nanoparticles. The kaolin nanocatalyst was dispersed in n-hexane to which titanium tetrachloride was added dropwise while continuously stirring for 2 h till milky white precipitate was obtained and then centrifuged to collect the powder. The prepared catalyst was characterised by different electron microscopic methods such as EXD, SEM, and TEM, which showed an average particle size of 50 nm. Its catalytic efficiency was evaluated through a one-stage preparation of a tautomeric product containing pyran moiety (**109**). The reaction proceeded through Knoevenagel condensation of aromatic aldehyde (**108**), sulphonyl acetonitrile (**107**) and hydroxy cyclohexadienone (**99**) (Figure 38). The rection further proceeds through Michael addition to form the tautomeric cyclisation. The use of Kaolin-infused titanium catalyst has dramatically reduced the reaction time to 3 h with excellent yields of 94% without considerable degeneracy in its efficiency for several consecutive cycles.

Rather et al. [92] have developed co-metal oxide-infused MWCNTs, using the impregnation method. The heterocatalyst was utilised for the preparation of chromeno-pyran analogues (**112**). Initially, MWCNTs were pre-treated with HNO_3_ and refluxed. The functionalised MWCNTs are washed several times and dried at 120 °C before subjecting them to sonication. Later, the MWCNTs were dispersed in water under sonication, followed by dropwise addition of the acidic solution of LaCl_3_ and CeCl_3_. This reaction mixture was continuously stirred at RT for 6 h till it formed a paste which was subjected to calcination at 300 °C for about 2 h to obtain La-Ce/MWCNTs. The catalyst was characterised using TEM, SEM, FTIR, XRD, EDX, TPD-NH_3_ and ICP-AES. The metallic load was 4.4% of La and 4.2% of Ce per gram of the catalyst. The metallic embeds appear as white dots on the CNT surface with an irregular texture giving it a nested morphology. A model one-pot reaction containing substituted aldehyde (**110**), hydroxy-coumarin (**16**), Meldrum acid (**111**) and nanocatalyst was explored. The process was presumed to be initiated through carbonyl group activation by La-Ce/MWCNTs resulting in the Knoevenagel condensation between the aldehyde and hyoxycoumarin, leading to the construction of 3-benzylidine-chroman-2,4-dione analogues (Figure 39). This reaction follows Michael addition with Meldrum acid and cyclisation to give rise to hemiketal, which is further dehydrated to form the final chromeno pyran analogues. A minimal amount of 100mg was sufficient to gain chromeno pyrans with excellent yields of 92% in a solvent-free single-pot reaction within 30 min at 70 °C.

Ghashang and co-workers [93] have studied the efficiency of MnO nanopowder in the preparation of thiochromeno pyran (**115**) and thiopyrano pyran (**117**) derivatives. The MnO nanoparticles were prepared by grinding the leaves of *Rosmarinus officialities* which were inserted into a gallon containing a solution of ethanol and water in 1:2 ratio, maintaining an alkaline medium using ammonia. The catalytic nanopowder was characterised by different techniques such as SEM and XRD, which suggest an average particle size of 73 nm with cubic morphology. A single-pot reaction system containing tetrahydrothiopyranone (**116**) or thiochromanone (**114**), benzaldehyde (**113**) and malononitrile (**3**) were investigated for the preparation of thiopyrano pyran and thiochromenopyran, respectively, to confirm the catalytic performance of MnO nanoparticles. The catalyst produced excellent yields (95%) under reflux in an ethanolic medium for 5 h reaction time (Figure 40). The benefit of the catalyst is its easy separation from the final product.

Moshtaghin and Zonoz [94] have constructed a magnetite-2H-phosphate (H_2_PO_4_@SCMNPs) to be used as a green heterogeneous catalyst for the preparation of 4*H*-benzopyran (**119**) analogues. Several instrumental techniques, such as FT-IR, SEM, TEM and XRD, revealed the inverse cubic spinel with a magnetite core covered with a silica shell adhered to dihydro phosphate moiety with a particle size ranging from 60–80 nm. The magnetic density on the nanoparticles was 23 emu/g. The catalytic activity was evaluated using a single-pot reaction system containing malononitrile (**3**), dimedone (**5**) and benzaldehyde (**118**) in the presence of 0.03 g of the catalyst. The protocol offered an excellent 90% yield within 15 min at 60 °C in a solvent-free environment (Figure 41).

Tavakol and Keshavarzipour [95] have fabricated magnetite nanoparticles (Mag-IPS-DES) furnished with urea-choline chloride in a eutectic solvent to be used as a green catalyst for synthesising tetrahydro pyran (**122**) moieties. The catalytic nanoparticles were characterised via FTIR, XRD, SEM and TEM techniques, reflecting a magnetic core with a spherical shape of 13 nm. A multicomponent reaction system containing aryl aldehyde (**120**), acetophenone (**121**) and malononitrile (**3**) in a single-pot was used for the preparation of tetrahydro pyran scaffolds in the presence of the catalytic particles (Figure 42). A 94% yield was obtained in 55 min reaction time with a 10 mg catalytic load. The green catalyst is durable with recyclability up to 5 cycles without degeneration.

Ramesh and co-workers [96] studied the catalytic affinity of Er(OTf)_3_ as an inorganic catalyst in the preparation of dihydropyran (**126**) moieties. An MCR system of butyl vinyl benzene (**123**) substituted butanoate (**125**) and aqueous formalin (**124**) was considered as a model reaction for the preparation of dihydropyrans (Figure 43). The liquid nature of the reactants makes it a comfortable solvent-free reaction system. The aryl components with both electron-withdrawing and donating groups gave excellent yields. Er(OTf)_3_ as a Lewis acid facilitates the Knoevenagel condensation and Hetero-Diels-Alder reactions. The catalyst gave remarkable gains of up to 95% with a catalytic load of 15 mg in 12 h of reaction time. Using inorganic compounds such as Lewis acids proves to be a green and highly efficient method for pyran analogues.

Aher and co-workers [97] have developed tungstenmolybdinum-quaternary vanado phosphoric acid, [H_5_PW_6_Mo_4_V_2_O_40_] encrypted on montmorillonite and used as a catalytic complex (nMont/VMWP) for generating tetrahydro pyrans (**128**) and polyhydroquinolines (**129**) under green conditions. Tungstenmolybdinum-quaternary vanado phosphoric acid was prepared by a hierarchy of acidified reactions and was characterised using FTIR, SEM-EDXA, TEM, SEM, TG-DA and XRD methods which indicated a 20% impregnation. The catalytic efficiency was examined for the single-stage preparation of pyran analogues from aryl aldehydes (**127**), dimedone (**5**) and active methylene (**3**) compounds. 20% of catalyst showed excellent catalytic activity with 85–95% yield in a short reaction time of 10 min under solvent-free conditions (Figure 44). The catalyst showed remarkable efficiency up to five cycles without losing its activity.

Alinezhad and co-workers [98] have formulated magnetic graphene particles functionalised with sulphonic acid (MGO-D-NH-(CH_2_)_4_-SO_3_H) to be used as a catalyst for the preparation of pyrano pirimidione (**133**) and tetrahydro pyran (**131**) analogues. The gathered catalytic particles characterised by FTIR, SEM, TEM, EDX and TGA, suggest that smooth sheet GO with wrinkled edges was modified into a rough surface due to the embedding of Fe_3_O_4_. Its pH was also evaluated using active-site titrations and found to be 2.1. The catalytic activity was assessed by using a single-pot reaction system consisting of aryl aldehyde (**130**) along with malononitrile (**3**) and dimethyl cyclohexanedione (**132**) for the preparation of tetrahydropyran and pyrano pyrimidione in the presence of the GO/Fe_3_O_4_ dentrimer sheets. The reaction gave a 98% yield in 3 min using a catalytic load of 0.2 g in an aqueous medium (Figure 45). The catalyst did not lose its efficiency for six consecutive cycles. The reaction mechanism involves a Knoevenagel condensation, Michael addition and cyclisation.

Maddila et al. [99] have synthesised arylsulphonyl pyran analogues (**136**) using fluorapatite infused with ruthenium (Ru/FAp) as a reusable catalyst. Fluorapatite was doped with ruthenium via the co-precipitation method, which was later dried and characterised using BET, BJH, P-XRD, TEM, SEM-EDX and SEM. The analysis suggests a bacillus structure with a particle size of 13–17 nm and a needle structure of 6–9 nm in size. The catalyst possessed surface area, pore volume and pore size of 44.1069 m^2^/g, 0.335 cm^3^/g, and 235.073, respectively. A multicomponent system consisting of substituted aromatic aldehydes (**134**), phenylsulphonyl acetonitrile (**135**) and dimedone (**5**) with nanoparticles in an ethanoic medium gave excellent yields (Figure 46). The probable reaction mechanism involved Knoevenagel condensation and Michael addition. The catalyst offered excellent product yield (98%) within 10 min using a load of 50 mg. This method helped produce superb results in a short reaction time under eco-friendly conditions.

Heravi and co-workers [100] have used a heteropolyacid (H_5_BW_12_O_40_) for the preparation of tetrahydropyran (**138**) via a multicomponent reaction system in a one-stage manner. The one-pot system containing malononitrile (**3**), dimedone (**5**) and aromatic aldehyde (**137**) in an equivalent amount plus catalyst in 50% ethanol was refluxed for 15 min (Figure 47). The possible reaction was anticipated via Knoevenagel, Michael addition and tautomeric cyclisation to produce the desired pyran moieties. The gathered products were characterised using various spectral techniques such as FTIR, ^1^HNMR and ^13^C NMR. The procedure gave 98% of yield in 15 min using a catalytic load of 10 mg. The catalyst was separated through evaporation and dried for one hour at 130 °C, which was reused for three consecutive cycles without marginal loss in efficiency.

## 3. Conclusions

Over the years, numerous innovative green synthetic methodologies have been developed for oxygen-containing heterocyclic molecules. In particular, pyran ring moieties have exhibited a wide range of biological applications and are contended with natural product-like scaffolds. One-pot synthetic methodologies of oxygen heterocyclic molecules under green conditions have attracted considerable attention in modern organic synthesis. This review emphasised the synthetic strategies for advancing oxygen-containing pyran heterocycles using different heterogeneous catalysts under green protocols. These pyran moieties were achieved from a multicomponent approach and focused on their catalytic role. Therefore, this review could be interesting to the readers of the development of biological potent oxygen heterocycles.

## Data Availability

Not applicable.

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
