# Peer review of "Recent Progress in the Multicomponent Synthesis of Pyran Derivatives by Sustainable Catalysts under Green Conditions"

_molecules, 2022, doi:10.3390/molecules27196347_

Round 1

Reviewer 1 Report

 In the manuscript entitled Recent progress in the multi-component synthesis of pyran derivatives by sustainable catalysts under green conditions from Sreekantha B. Jonnalagadda and co-workers summarized the recent development of pyran derivatives by the sustainable catalyst and green conditions. This manuscript could be helpful to medicinal and sustainable chemistry researchers and interested Molecules readers. However, to improve the quality of the manuscript, the authors need to address the below comments.

 In 4H-pyran; H must be italic. Authors need to correct the entire manuscript.  R groups in all chemDraws random. Need to provide uniform in chronological order (EDG to EWG or EWG to EDG).

The author's cloud includes the reaction time in the ChemDraw.  

Aldehyde has many numbers in the manuscript. Therefore, it must be one number.

The authors mentioned the characterization data of the catalyst and 4H-pyran derivatives in the manuscript. However, instead of it, the discussion of the trends of the yields in the substrate scope or several attempts to optimize the yields could improve the value of the manuscript.  

Authors could include/cite the below papers in the manuscript in a suitable place.

a). Recyclable Bi2WO6-nanoparticle mediated one-pot multicomponent reactions in aqueous medium at room temperature RSC Adv., 2014,4, 54168-54174.

b).  Synthesis of functionalized chromene and spirochromenes using l-proline-melamine as highly efficient and recyclable homogeneous catalyst at room temperature Tetrahedron Lett. 2017, 58, 42004204. 

Reviewer 2 Report

The authors (S. Maddila, et al.) described the multi-component synthesis of pyran derivatives by mainly using heterogeneous catalysts. This manuscript was well organized and included a lot of references. I think this paper would give a good effect for many chemists, so it would be acceptable for Molecule as the present format. There are some small comments below.

1.        H of ‘4H-pyran’ should be shown by italic. – e.g. p.2 l.66 and l.77, p.17 l.455 and l.457, and others should be checked.

2.        The notation of reaction conditions such as solvents and temperatures in Schemes should be unified. – e.g. ‘EtOH, reflux’, ‘EtOH/reflux’, ‘C2H5OH; Reflux’, ‘EtOH/RT’, or others are mixed.

3.        Similar to the above, the notation of substituents in Schemes should be unified. – e.g. ‘2,4-Di-Cl’, ‘2,4-di-Cl’, ‘2,4-Cl2’, or others are mixed.

4.        P.11 l.290; ‘The efficacy of the nanocomposites in the preparation of pyrano[2,3-c]-chromenes (50).’ is what?

5.        P.13 l.338; H1 and C13 NMR should be corrected to 1H and 13C.

6.        P.14 l.365; Brønsted acid.

7.        Scheme 23; compound 38 should be used middle dot.

8.        P.15 l.404; Fe2+/Fe3+.

9.        P.15 l.405; The font of degree should be checked. It seems to be using a different font.

10.    P.16 l.422; CNTS should be corrected to CNTs.

11.    P.20 l.540; copper.

12.    P.20 l.616 and l.624; the time should be unified to h or hrs. And others should be checked.

13.    P.28 l.733; Å?

Round 2

Reviewer 1 Report

In the updated manuscript all of my suggestions and comments have been addressed sufficiently. The manuscript is publishable in its current form.